# Thymosin Beta 4 Protects Hippocampal Neuronal Cells against PrP (106–126) via Neurotrophic Factor Signaling

**DOI:** 10.3390/molecules28093920

**Published:** 2023-05-06

**Authors:** Sokho Kim, Jihye Choi, Jungkee Kwon

**Affiliations:** 1Department of Laboratory Animal Medicine, College of Veterinary Medicine, Jeonbuk National University, Gobong-ro 79, Iksan 54596, Jeollabuk-do, Republic of Korea; raios@hanmail.net (S.K.); jyyye@naver.com (J.C.); 2Knotus Co., Ltd., Incheon 22014, Republic of Korea

**Keywords:** thymosin beta 4, prion, neurotrophic factor, NGFRp75, TrkA, TrkB

## Abstract

Prion protein peptide (PrP) has demonstrated neurotoxicity in brain cells, resulting in the progression of prion diseases with spongiform degenerative, amyloidogenic, and aggregative properties. Thymosin beta 4 (Tβ_4_) plays a role in the nervous system and may be related to motility, axonal enlargement, differentiation, neurite outgrowth, and proliferation. However, no studies about the effects of Tβ_4_ on prion disease have been performed yet. In the present study, we investigated the protective effect of Tβ_4_ against synthetic PrP (106–126) and considered possible mechanisms. Hippocampal neuronal HT22 cells were treated with Tβ_4_ and PrP (106–126) for 24 h. Tβ_4_ significantly reversed cell viability and reactive oxidative species (ROS) affected by PrP (106–126). Apoptotic proteins induced by PrP (106–126) were reduced by Tβ_4_. Interestingly, a balance of neurotrophic factors (nerve growth factor and brain-derived neurotrophic factor) and receptors (nerve growth factor receptor p75, tropomyosin related kinase A and B) were competitively maintained by Tβ_4_ through receptors reacting to PrP (106–126). Our results demonstrate that Tβ_4_ protects neuronal cells against PrP (106–126) neurotoxicity via the interaction of neurotrophic factors/receptors.

## 1. Introduction

Prion disease is the common name for transmissible spongiform encephalopathies (TSE), neurodegenerative diseases of animals and humans [1]. These diseases are sporadic of inherited in origin and are characterized by histopathology involving spongiform degeneration with neuronal loss and gliosis [2]. Although the etiology of prion disease has not been well elucidated, major evidence suggests that modification of prion protein (PrP) from a normal cellular protein (PrP^c^) to a disease-specific species called the pathological scrapie isoform (PrP^sc^) causes insolubility and protease resistance, resulting in the disruption of neuronal homeostasis. The PrP fragment (106–126) has a similar function as PrP^SC^ and easily aggregates in brain cells, causing resistance to proteolytic enzymes [3]. To study the prions involved in neurodegenerative diseases, model agents such as synthetic peptides homologous to PrP have been used (106–126) [4]. Prion peptide PrP (106–126) has been demonstrated to be neurotoxic in brain cells due to its spongiform degenerative, amlyoidogenic, and aggregative propertied both in vivo and in vitro [5]. Prion-related encephalopathies are rare and deadly diseases caused by the abnormal transition of normal cellular prion protein into a pathogenic protease-resistant form. Synthetic peptides similar to this pathogenic protein, such as PrP (106–126), have been used to study the mechnisms of neurodegeneration. Both full-length PrP^SC^ and PrP (106–126) have been shown to be toxic to neurons, and various mechanisms have been proposed to explain neuronal death in prion diseases [3,5].

Neurotrophic factors called neurotrophins are related to neurogenesis and are important for neuronal survival in the brain [6]. A previous study has suggested that neurotrophic factors can be used as therapeutic agents in neuronal disorders [7]. Neurotrophic factors are a family of proteins with similar structure and function to nerve growth factor (NGF), brain-derived neurotrophic factor (BDNF), and glial cell line-derived neurotrophic factor (GDNF) [8]. The actions of neurotrophic factors are mediated by two membrane receptor signaling systems, nerve growth factor receptor p75 (NGFRp75) and the tropomyosin related kinase (Trk) family including TrkA and TrkB [9]. Each neurotrophic factor reveals a different binding specificity for specific receptors. NGF preferentially binds to TrkA, a high-affinity nerve growth factor receptor, and NGFRp75, a low-affinity nerve growth factor receptor [10]. BDNF preferentially binds to TrkB. A previous study demonstrated that the relationship between neurotrophic factors and their receptor is involved in the pathogenesis of neurodegenerative diseases [11]. With prion diseases, PrP (106–126)-induced apoptosis of mouse neuronal cells reacted to the NGFRp75 signaling pathway [12], suggesting that NGFRp75 might be particularly related to the pathogenesis of prion diseases.

Thymosin beta 4 (Tβ_4_) is a peptide identified as an actin monomer binding molecule present in all mammalian species [13]. Tβ_4_ plays a role in many cellular processes, including motility, axonal enlargement, differentiation, neurite outgrowth, and proliferation [14]. Based on the above roles of Tβ_4_, the physiological and pathological nervous system processes mediated by Tβ_4_ have been established [15]. The underlying mechanism of Tβ_4_ in the nervous system may be related to its neuronal growth effects [16]. The previous study, we found Tβ_4_ prevent neurodegenerative diseases caused by PrP (106–126)-induced blood–brain barrier (BBB) dysfunction [17]. In addition, Tβ_4_ can regulate autophagy activation not only in PrP (106–126)-induced HT22 cells [18], but also in LPS and ATP-induced RAW264.7 and LX-2 cells [19]. Tβ_4_ has also demonstrated neuroprotective and neurorestorative potential within various neurological injury models [20]. In particular, the neuroprotective effects of Tβ4 have been observed in a mouse model of neuroinflammatory BBB dysfunction induced by systemic infection with LPS [21,22]. However, since Tβ_4_ cannot directly pass through the BBB [20], it is believed that these effects are mediated outside of the central nervous system, affecting the body as a whole. However, no direct evidence of neurotrophic factors or neurotrophic receptor involved with Tβ_4_ has been suggested yet. The present study examined direct interaction among Tβ_4_, neurotrophic factors, and neurotrophic receptors in the presence or absence of PrP (106–126). Accordingly, neuronal cell protection by Tβ_4_ against PrP (106–126) neurotoxicity via neurotrophic signaling pathways was assessed. The results of the present study suggest the first evidence of an interaction between Tβ_4_ and PrP (106–126) via possible underlying mechanisms involved in neurotrophic factors/receptors. These results may lead to a novel therapeutic strategy for treating prion diseases.

## 2. Results

### 2.1. Tβ_4_ Protects HT22 Cells against PrP (106–126)

To evaluate the effect of Tβ_4_ on PrP (106–126)-treated HT22 cells, cell viability and ROS activity were confirmed. As shown in Figure 1A, Tβ_4_ increased cell viability in a dose-dependent manner. Figure 1B shows that synthetic PrP (106–126) decreased cell viability in a dose-dependent manner compared to scrambled PrP (106–126), which was used as a positive control for synthetic peptide toxicity. A significant reduction was shown over 100 μM PrP (106–126) treatment. Accordingly, the effect of Tβ_4_ on 100 μM PrP (106–126) treated HT22 cells was assessed. As shown in Figure 1C, over 400 ng/mL of Tβ_4_ on 100 μM PrP (106–126)-treated cells resulted in cell viability similar to non-treated control. ROS activity was significantly reduced by Tβ_4_ treatment in dose-dependent manner on HT22 cells treated with 100 μM PrP (106–126) (Figure 1D). Based on these results, adequate doses of Tβ_4_ (400 ng/mL) and PrP (106–126) (100 μM) were used for subsequent experiments.

### 2.2. Tβ_4_ Inhibited Apoptosis Induced by PrP (106–126)

The cellular toxicity of PrP (106–126) directly induced apoptosis such as Bcl-family, Bax, cleaved caspase-3. Furthermore, the anti-apoptosis effect of Tβ_4_ on PrP (106–126) treated HT22 cell was confirmed. As shown in Figure 2, the anti-apoptosis protein Bcl-xL was increased by Tβ_4_ while it was decreased by PrP (106–126). Apoptosis proteins such as Bax and cleaved caspase-3 were decreased by Tβ_4_ while they were increased by PrP (106–126). These results suggest that Tβ_4_ inhibited apoptosis induced by PrP (106–126) in HT22 cells.

### 2.3. Tβ_4_ Induced Neruotrophic Factors Such as NGF and BDNF

The above results suggest that Tβ_4_ improved neuronal cell survival. To dissect the possible mechanisms of Tβ_4_, neurotrophic factor was confirmed to affect the cell physiology of neurons (Figure 3). As expected, Tβ_4_ revealed significant results on both RNA and protein levels of neurotrophic factors. PrP (106–126) decreased the levels of NGF and BDNF in both RNA and protein. Only Tβ_4_ treated cells showed significant increase in both NGF and BDNF in RNA and protein levels. Tβ_4_ increased both NGF and BDNF compared to reduction by PrP (106–126) in HT22 cells. Thus, Tβ_4_ induced NGF and BDNF to improve neuronal cell survival.

### 2.4. Intrinsic Tβ_4_ Induced Neurotrophic Factors and Its Own Receptors

As shown in Figure 3, the effect of intrinsic Tβ_4_ on neurotrophic factors and its own receptors was confirmed through Tβ_4_ siRNA. As shown in Figure 4A, RNA levels were changed by Tβ_4_ siRNA. Tβ_4_, NGF, and BDNF expression showed similar tendencies as each other. Tβ_4_ siRNA significantly inhibited Tβ_4_ levels as well as NGF and BDNF levels. Co-treatment with Tβ_4_ on Tβ_4_-siRNA-treated cells revealed the reverse effect of Tβ_4_ siRNA. The protein levels of Tβ_4_, NGF and BDNF showed similar results as RNA levels of Tβ_4_, NGF, and BDNF (Figure 4B). Additionally, altered receptors were confirmed as a reaction to neurotrophic factors. As shown in Figure 4C, Tβ_4_ reduced NGFRp75, possibility due to reaction with PrP, while it induced both TrkA and TrkB due to reaction with NGF and BDNF. PrP (106–126) significantly increased NGFRp75, which was boosted with Tβ_4_ siRNA. The deletion of Tβ_4_ by Tβ_4_ siRNA revealed a similar tendency as RNA levels of PrP (106–126)-treated cells. Tβ_4_ treated with PrP (106–126) and Tβ_4_ siRNA resulted in reversed RNA levels of Tβ_4_, NGFRp75, TrkA, and TrkB compared to cells co-treated with PrP (106–126) and Tβ_4_ siRNA. The protein levels of Tβ_4_, NGFRp75, TrkA, and TrkB were confirmed to belong to the same groups as RNA results. As shown in Figure 4D, protein levels of Tβ_4_, NGFRp75, TrkA, and TrkB revealed similar tendencies as RNA levels. These results suggested that both intrinsic and extrinsic Tβ_4_ affected NGF. In addition, BDNF has its own receptors, such as NGFRp75, TrkA, and TrkB.

Moreover, both TrkA inhibitor and TrkB inhibitor were used to artificially delete each receptor in Tβ_4_ treated cells (Figure 5). Each inhibitor significantly reduced the phosphorylated form of each protein level. Tβ_4_ increased the total form of TrkA and TrkB expression as well as the phosphorylated form of TrkA and TrkB. TrkA inhibitor and TrkB inhibitor significantly reduced the phosphorylated form induced by Tβ_4_.

### 2.5. Tβ_4_ Protects HT22 Cells from PrP (106–126) via Induced Neurotrophic Factors

Finally, repeated experiments were performed to confirm results regarding apoptosis, cell viability, and ROS. As shown in Figure 6A, Tβ_4_ inhibited cleaved caspase-3, which was reversed by Tβ_4_ siRNA, PrP (106–126), TrkA inhibitor, and TrkB inhibitor. Accordingly, Tβ_4_ increased the cell viability that was reversed by Tβ_4_ siRNA, PrP (106–126), TrkA inhibitor, and TrkB inhibitor (Figure 6B). ROS activity also had the same tendency as cleaved caspase-3 results (Figure 6C). These results suggested that Tβ_4_ protects hippocampal neuronal cells against PrP (106–126) via an induced neurotrophic factor and reaction with its receptors.

## 3. Discussion

The present study demonstrated that Tβ_4_ protects hippocampal neuronal cells against PrP (106–126) via upregulation of neurotrophic factors and their receptors. The mammalian nervous system naturally produces Tβ_4_ during postnatal development [15]. Moreover, early-stage embryogenesis involves abundant expression of Tβ_4_ in neural tissue [23]. Tβ_4_ is distributed in the adult forebrain including the cerebral cortex, hippocampal entorhinal region, cerebellum, infundibular region, substantia nigra pars compacta, supraoptic, medial amygdaloid, and dorsal premammillary nuclei [24,25]. In a previous study, neuron and glial cells induced Tβ_4_ with focal brain ischemia and kainic acid treatment [26]. The previous study demonstrated that Tβ_4_ has a crucial role in physiological and pathological process in the nervous system. Extrinsic Tβ_4_ treatment is also involved in motility, axon growth, and synapse generation in neurons after brain damage [27]. A reasonable proposed mechanism for this is neurotrophic effects [16]. However, any direct evidence for the relationship between Tβ_4_ and neurotrophic factors has not yet been elucidated, although NGF has been shown to induced Tβ_4_ in PC12 cells [28]. Thus, we studied how Tβ_4_ involved with both neurotrophic factors and their receptors to protect against PrP toxicity.

Prion disease, also known as transmissible spongiform encephalopathy, is the only natural occurring infectious protein misfolding disease. It is caused by the PrP^C^ into PrP^SC^, resulting in the accumulation of misfolded protein particels [29,30]. Numerous studies have been conducted to identify effective agents for drug development in prion diseases [30]. This study used PrP (106–126) as a derivative to simulate the pathological signaling observed in prion diseases. PrP (106–126), a synthetic peptide homologous to PrP (106–126), was widely used as an extrinsic treatment to study prion disease [4]. PrP (106–126) induced neurotoxicity in neuronal cells due to its amyloidogenic properties both in vivo and in vitro [5]. In particular, in our previous study, we demonstrated the effectiveness of Tβ_4_ on neurotoxicity and on autophagy activity in PrP (106–126)-induced HT22 cells [18]. Results in Figure 1 showed that Tβ_4_ protects hippocampal neuronal HT22 cells against PrP (106–126), demonstrated by increased cell viability and reduced ROS activity. Moreover, the PrP (106–126)-induced proteins associated with apoptosis were reversed by Tβ_4_ (Figure 2). To dissect reasonable mechanisms for this, neurotrophic factors such as NGF and BDNF were examined (Figure 3). Both the RNA and protein levels of NGF and BDNF were increased by Tβ_4_ compared to reduced NGF and BDNF with PrP (106–126).

Neurotrophic factors and their receptor have a known role in neurogenesis and protection in mammalian nervous systems. Neurotrophic factor mechanisms are related to throsin kinase receptors such as TrkA, TrkB, TrkC and NGFRp75, a subfamily of the tumor necrosis factor receptor [9]. Different neurotrophic factors have binding specificities for precise receptors. However, these interactions can be altered with regulation by receptor dimerization, structural modifications, or association with NGFRp75. The interaction between neurotrophic factors and their receptors is related to neurodegenerative diseases [11]. In prion diseases, PrP (106–126)-induced apoptosis in neuroblastoma cells involves up-regulated NGFRp75 and the nuclear factor kappa B (NF-κB) signaling pathway [12]. Based on previous studies, neurotrophic factors binding to their receptors affect the pathogenesis of prion diseases. Thus, the interaction between Tβ_4_ and neurotrophic factor/receptors was examined (Figure 4). Deletion of Tβ_4_ inhibited expression of NGF and BDNF, as well as their high affinity receptors TrkA and TrkB (Figure 4B,D). NGFRp75, which reacts to PrP (106–126), was incrased by Tβ_4_ siRNA. PrP (106–126) accelerated the effect of Tβ_4_ siRNA on NGFPp75 and reduction in TrkA and TrkB (Figure 4C,D). Thus, NGF and BDNF and their receptors have important roles in the protective effect of intrinsic Tβ_4_ against PrP (106–126).

Nerve growth factor (NGF) binds to TrkA, and BDNF and neurotrophin 4 bind to TrkB. However, the low affinity of binding of NGF to TrkA and binding of BDNF to TrkB can be transformed by dimerization of receptors, structural deformation, and association with p75NTR receptors, which can also increase ligand selectivity [31]. To confirm the interaction between Tβ_4_ and neurotrophic receptors, TrkA inhibitor and TrkB inhibitor were used. As shown in Figure 5, each particular inhibitor inhibited the phosphorylated form of TrkA and TrkB. Tβ_4_ significantly reversed the expression of phosphorylated TrkA and TrkB caused by each inhibitor. This result indicated that Tβ_4_ induced neurotrophic receptors for neuron survival. All corresponding treatments with Tβ_4_ in HT22 cells revealed coincident results that Tβ_4_ significantly reduced the neurotoxicity induced by PrP (106–126) via interaction of neurotrophic factors/receptors (Figure 6). In addition, the accumulation of misfolded prions leads to vesicle stress and disturbances in calcium signaling regulation, which can cause mitochondrial dysfunction, compounding the stress produced by misfolded proteins [32]. ROS production due to intracellular oxidative stress affects the prion infection process, contributing to apoptosis and damage [33]. Tβ_4_ decreased apoptosis (Figure 6A) and ROS activity (Figure 6C).

The present results show for the first time that Tβ_4_ reduces neuronal cell toxicity induced by PrP (106–126), regarded as a cause of prion disease. Tβ_4_ also plays a crucial role as a key recovery factor that induces NGF and BDNF, signals transmitted to TrkA and TrkB in the neuronal survival signaling pathway (Figure 7). Based on these findings, we suggest that Tβ_4_ could serve as a novel therapeutic strategy for treating prion disease via the neurotrophic factor/receptor signaling pathway. This interaction warrants further investigation regarding its role in neurotrophic factor balance.

## 4. Materials and Methods

### 4.1. Chemicals

Tβ_4_ was purchased from Tocris Bioscience (Bristol, UK). TrkA inhibitor was purchased from Santa Cruz Biotechnology (Dallas, TX, USA). TrkB inhibitor was purchased from Sigma-Aldrich (St. Louis, MO, USA). Primary antibodies for NGF, BDNF, Bcl-xL, Bax, Cleaved caspase-3, Caspase-3, Thymosin beta 4 (Tβ_4_), NGFRp75, phosphorylated (p)-TrkA, TrkA, p-TrkB, TrkB, and β-actin were purchased from Abcam (Cambridge, UK). Secondary antibodies (i.e., anti-rabbit, anti-goat, or anti-mouse IgG antibody conjugated with horseradish peroxidase) was obtained from Thermo (Temecula, CA, USA). Synthetic PrP (106–126) (KTNMKHMAGAAAAGAVVGGLG) and scrambled PrP (106–126) (NGAKALMGGHGAYKVMVGAAA) were synthesized by Peptron (Seoul, Korea). PrP peptides were dissolved in sterile dimethyl sulfoxide at a concentration of 10 mM and stored at −72 °C. All other chemicals and reagents were of analytic grade.

### 4.2. Cell Culture

The hippocampal neuronal cell line (HT22; ATCC, Rockville, MD, USA) originated from mouse was maintained in Dulbecco’s modified Eagle’s medium (DMEM; HyClone, Logan, UT, USA) containing 10% fetal bovine serum (FBS; Hyclone, Canada) and 1% penicillin–streptomycin (Sigma-Aldrich, St. Louis, MO, USA) in a 37 °C humidified incubator with 5% CO_2_. The medium was changed every 2~3 days. Cells were treated with PrP (106–126) at 100 μM and Tβ_4_ at 400 ng/mL. TrkA inhibitor at 3 μM and TrkB inhibitor at 5 μM were used.

### 4.3. Small Interfering RNA (siRNA) Rransfection

HT22 cells were plated in 6 cm^2^ dishes (4 × 10^5^ cells/dish) until confluence and then starved for 24 h. The DharmaFECT 1 small interfering RNA (siRNA) Transfection Reagent (Dharmacon, Denver, CO, USA) was used to transfect cells with 50 nM Tβ_4_ siRNA or scrambled siRNA oligonucleotides (Dharmacon, Danver, CO, USA) according to the manufacturer’s instructions, as reported previously.

### 4.4. Cell Viability

Cell viability was determined using a 3-(4,5-Dimethylthiazol-2-yl)-2,5-diphenyltetrazolium bromide (MTT) assay kit from Sigma-Aldrich (St. Louis, MO, USA) according to the manufacturer’s instructions. HT22 cells were grown on 96-well plates (SPL, Pochon, Korea) at a density of 2 × 10^4^ cells/well. After the corresponding treatment, cell viability was evaluated by assaying the ability of functional mitochondria to catalyze reduction in MTT to a formazan salt by mitochondrial dehydrogenases. The index of cell viability was determined with multiplate reader spectrophotometry (PowerWave 2, Bio-Tek Instruments, Winooski, VT, USA) based on an absorbance of 570 nm.

### 4.5. Intracellular Reactive Oxygen Species Assay

The level of intracellular reactive oxygen species (ROS) was quantified by fluorescence using 2′,7′-dichlorodihydrofluorescein diacetate (DCF-DA; Invitrogen, Carlsbad, CA, USA). Cells were grown on 48-well plates and incubated in corresponding treatment conditions for 6 h. After the incubation period, cells were washed with phosphate-buffered saline (PBS) and stained with DCF-DA in PBS for 30 min in the dark. Cells were then washed twice with PBS and extracted with 0.1% Tween-20 in PBS for 10 min at 37 °C. Fluorescence was recorded using an excitation wavelength of 490 nm and emission wavelength of 525 nm.

### 4.6. RNA Preparation and Real-Time (RT)-PCR

Total RNA was isolated from cells and precipitated with Ribo EX (Geneall, Daejeon, South Korea) according to the manufacturer’s instructions. mRNA was reverse transcribed to cDNA using a Maxime RT PreMix kit (Intron, Seongnam, South Korea) according to the manufacturer’s instructions. For real-time RT-PCR, cDNA was amplified using a Mastercycler Gradient 5331 Thermal Cycler (Eppendorf, Germany). An ABI Step One Plus Sequence Detection System (Applied Biosystems, Middlesex County, MA, USA) was used to monitor real-time PCR runs by measuring fluorescence signals after each cycle. Specific primers for each gene were designed using Primer Express software (Applied Biosystems). The following forward and reverse primers were used for real-time RT-PCR quantification (forward and reverse): 5′-AAACCCGATATGGCTGAGATTG -3′ and 5′-GCCTGCTTGCTTCTCCTGTT-3′ for Tβ_4_, 5′-TGGGCTTCAGGGACAGAGTC-3′ and 5′-CAGCTTTCTATACTGGCCGCAG-3′ for NGF, 5′-AACCATAAGGACGCGGACTT-3′ and 5′-TGCAGTCTTTTTATCTGCCG-3′ for BDNF, 5′-ACCATCTCAGGCCTTTCCTT-3′ and 5′-TGTTGGGTG GCCTAGGTTAG-3′ for NGFRp75, 5′-GAGGAGCAAATTTGGGATCA-3′ and 5′-GGTGCAGACTCCAAAGAAGC-3′ for TrkA, 5′-AAGGACTTTCATCGGGAAGCTG-3′ and 5′- TCGCCCTCCACACAGACAC-3′ for TrkB, and 5′-TGTGTCCGTCGTGGATCTGA-3′ and 5′-CAACACCTCAACAGGAGTGGACA-3′ for glyceraldehyde-3-phosphate dehydrogenase (GAPDH), the housekeeping gene used as an internal control. All experiments were performed at least three times. Data were normalized to the reference gene, GAPDH.

### 4.7. Immunoblotting Analysis

Total proteins were extracted with a RIPA lysis buffer with EDTA containing a protease inhibitor cocktail and a phosphatase inhibitor cocktail. Proteins in cells were subjected to sodium dodecyl sulfate–polyacrylamide gel electrophoresis using 8%, 10%, and 14% gels, and then were electrophoretically transferred to polyvinylidene fluoride (PVDF) membranes (#162177, Bio-Rad, Contra Costa, CA, USA). The membranes were blocked with 5% skim milk in PBS and then individually incubated with each primary antibody diluted to 1:1000 in 1% skim milk in PBS overnight at 4 °C. Blots were further incubated with each secondary antibody diluted to 1:10,000 at room temperature for 1 h. The immunoreactions were visualized using SuperSignal West Dura Extended Duration Substrate (Thermo Fischer Scientific, San Jose, CA, USA) and analyzed using a ChemiImager system (Alpha Innotech, San Leandro, CA, USA).

### 4.8. Statistical Analysis

The data were analyzed using Student’s *t*-test (for two groups), one-way ANOVA, and Tukey’s test (for more than two groups). Data are presented as mean and SEM values. The cutoff for statistical significance was set at *p* < 0.05. All analyses were performed using the Statistical Package for Social Sciences (version 13.0 for Windows, SPSS, Chicago, IL, USA).

## 5. Conclusions

In conclusion, our study provides evidence that thymosin beta 4 (Tβ_4_) has a protective effect against neurotoxicity induced by synthetic prion protein peptide (PrP) (106–126) in hippocampal neuronal cells. Tβ_4_ treatment significantly improved cell viability, reduced ROS levels, and decreased apoptotic protein expression induced by PrP (106–126). Additionally, Tβ_4_ appears to play a role in maintaining a balance of neurotrophic factors and receptors, which are essential for proper signaling in the nervous system. Although there is promising in vitro evidence supporting the effectiveness of Tβ4, it is important to note that there is a lack of in vivo experiments to confirm these findings. Therefore, further research, particularly in vivo studies, is needed to fully evaluate the potential of Tβ4 as a therapeutic agent. Our findings suggest that Tβ_4_ may have therapeutic potential in the treatment of prion diseases and other neurodegenerative disorders.

## Figures and Tables

**Figure 1 molecules-28-03920-f001:**
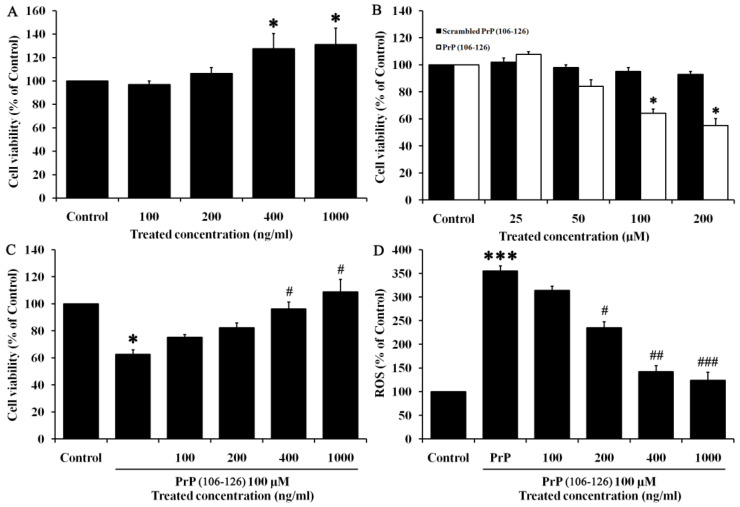
Cell viability and ROS activity in Tβ_4_ and PrP (106–126)-treated cells. Cell vability was assessed by MTT assay. (**A**) Tβ_4_ was treated dose-dependently (100~1000 ng/mL) for 24 h. (**B**) PrP (106–126) and scrambled PrP (106–126) were treated dose dependently (25–200 μM) for 24 h. (**C**) Tβ4 was treated dose-dependently (100–1000 ng/mL) with PrP (106–126) at 100 μM for 24 h. ROS activity was assessed by DCF-DA staining. (**D**) Tβ4 was treated dose dependently (100–1000 ng/mL) with PrP (106–126) at 100 μM for 24 h. Data are represented as mean ± SEM (n = 3). * *p* < 0.05, compared with control. *** *p* < 0.001, compared with control. # *p* < 0.05, compared with only PrP (106–126)-treated group. ## *p* < 0.01, compared with only PrP (106–126)-treated group. ### *p* < 0.001, compared with only PrP (106–126)-treated group.

**Figure 2 molecules-28-03920-f002:**
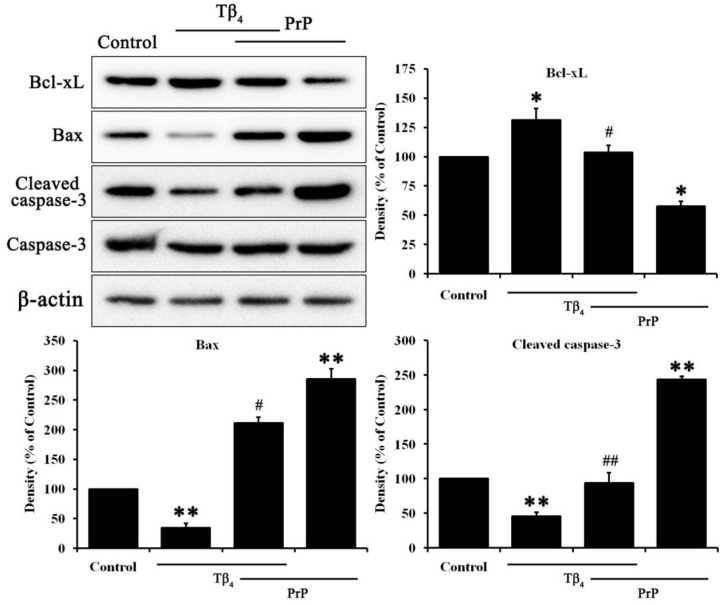
Apoptosis in Tβ_4_ and PrP (106–126)-treated cells. Tβ_4_ at 400 ng/mL treated with or without PrP (106–126) at 100 μM for 24 h. Bcl-xL, Bax, cleaved caspase-3, caspase-3, and β-actin were confirmed by immunoblotting. Data are represented as mean ± SEM (n = 3). * *p* < 0.05, compared with control. ** *p* < 0.01, compared with control. # *p* < 0.05, compared with PrP (106–126)-treated group. ## *p* < 0.01, compared with only PrP (106–126)-treated group.

**Figure 3 molecules-28-03920-f003:**
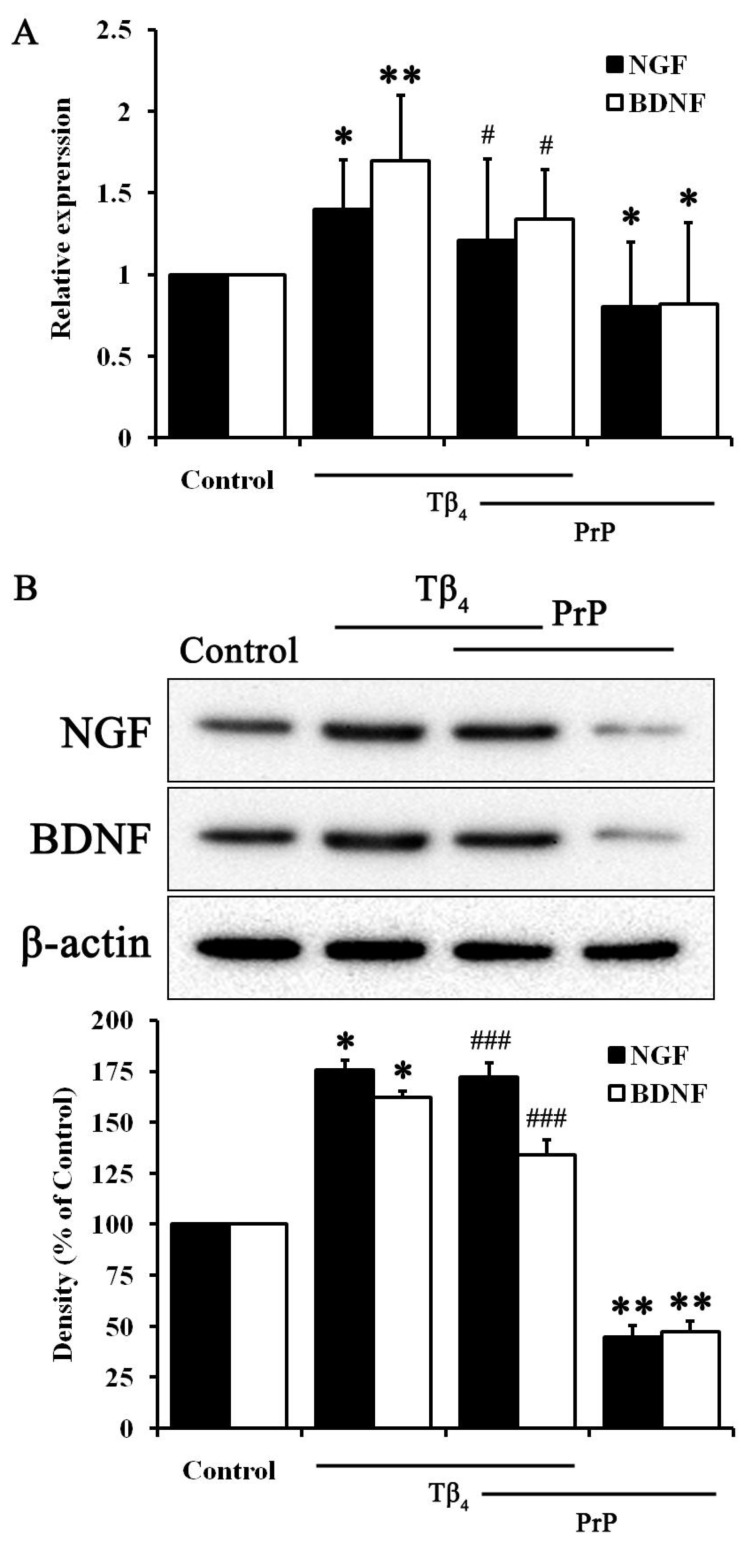
NGF and BDNF expression in Tβ_4_- and PrP (106–126)-treated cells. Tβ_4_ at 400 ng/mL treated with or without PrP (106–126) at 100 μM for 24 h. (**A**) RNA levels of NGF and BDNF. (**B**) Protein levels of NGF and BDNF. Data are represented as mean ± SEM (n = 3). ** p* < 0.05, compared with control. *** p* < 0.01, compared with control. *# p* < 0.05, compared with only PrP-treated group. *### p* < 0.001, compared with only PrP-treated group.

**Figure 4 molecules-28-03920-f004:**
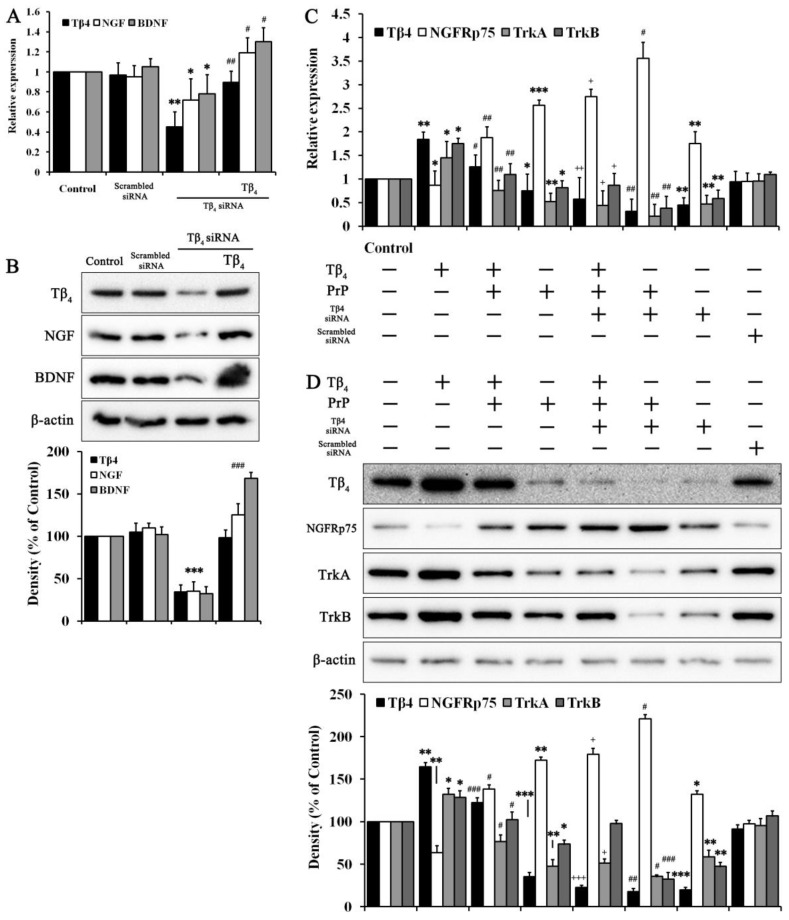
Neurotrophic factors and their receptors affected by Tβ_4_. (**A**) RNA levels and (**B**) protein levels of Tβ_4_, NGF, and BDNF in 400 ng/mL Tβ_4_ treated with or without scrambled siRNA and Tβ_4_ siRNA for 24 h. Data are represented as mean ± SEM (n = 3). ** p* < 0.05, compared with control. *** p* < 0.01, compared with control. **** p* < 0.001, compared with control. *# p* < 0.05, compared with Tβ_4_ siRNA group. *## p* < 0.01, compared with Tβ_4_ siRNA group. *### p* < 0.001, compared with Tβ_4_ siRNA group. (**C**) RNA levels and (**D**) protein levels in 400 ng/mL Tβ_4_ with or without scrambled siRNA, Tβ_4_ siRNA, and PrP (106–126) at 100 μM for 24 h. Data are represented as mean ± SEM (n = 3). ** p* < 0.05, compared with control. *** p* < 0.01, compared with control. **** p* < 0.001, compared with control. *# p* < 0.05, compared with only PrP-treated group. *## p* < 0.01, compared with only PrP-treated group. *### p* < 0.001, compared with only PrP-treated group. + *p* < 0.05, compared with Tβ_4_ + PrP-treated group. ++ *p* < 0.01, compared with Tβ_4_ + PrP-treated group. +++ *p* < 0.001, compared with Tβ_4_ + PrP-treated group.

**Figure 5 molecules-28-03920-f005:**
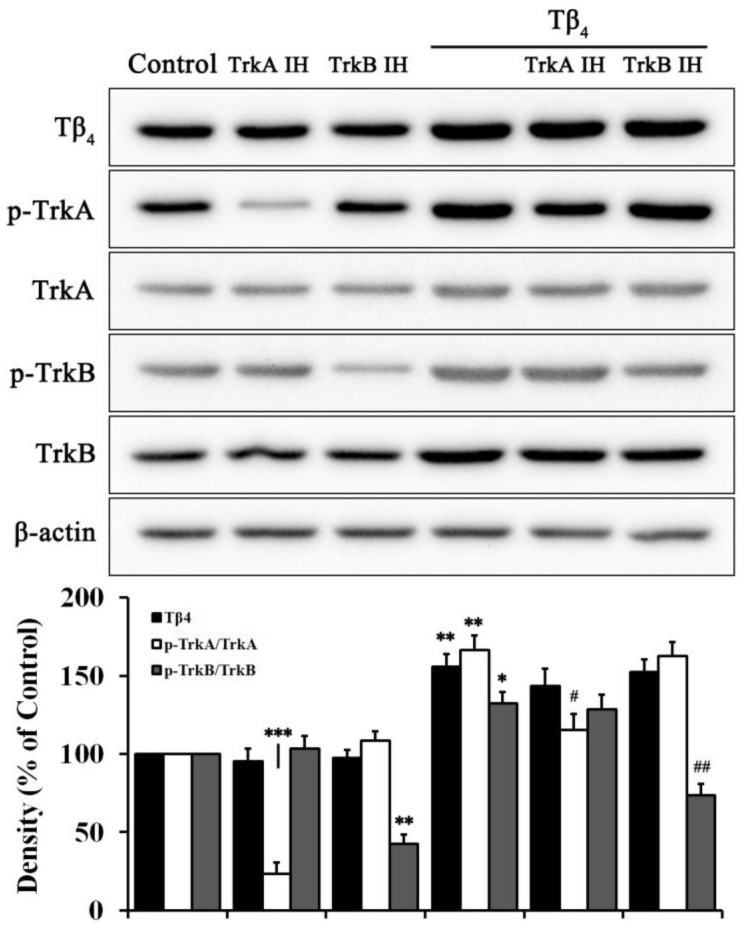
Relationship between Tβ_4_ and neurotrophic factor receptors. Protein levels of Tβ_4_, p-TrkA, TrkA, p-TrkB, TrkB, and β-actin in 400 ng/mL Tβ_4_ treated with or without 5 μM TrkA inhibitor and 20 μM TrkB inhibitor for 24 h. Data are represented as mean ± SEM (n = 3). ** p* < 0.05, compared with control. *** p* < 0.01, compared with control. **** p* < 0.001, compared with control. *# p* < 0.05, compared with only Tβ4-treated group. *## p* < 0.01, compared with the Tβ4-treated only group.

**Figure 6 molecules-28-03920-f006:**
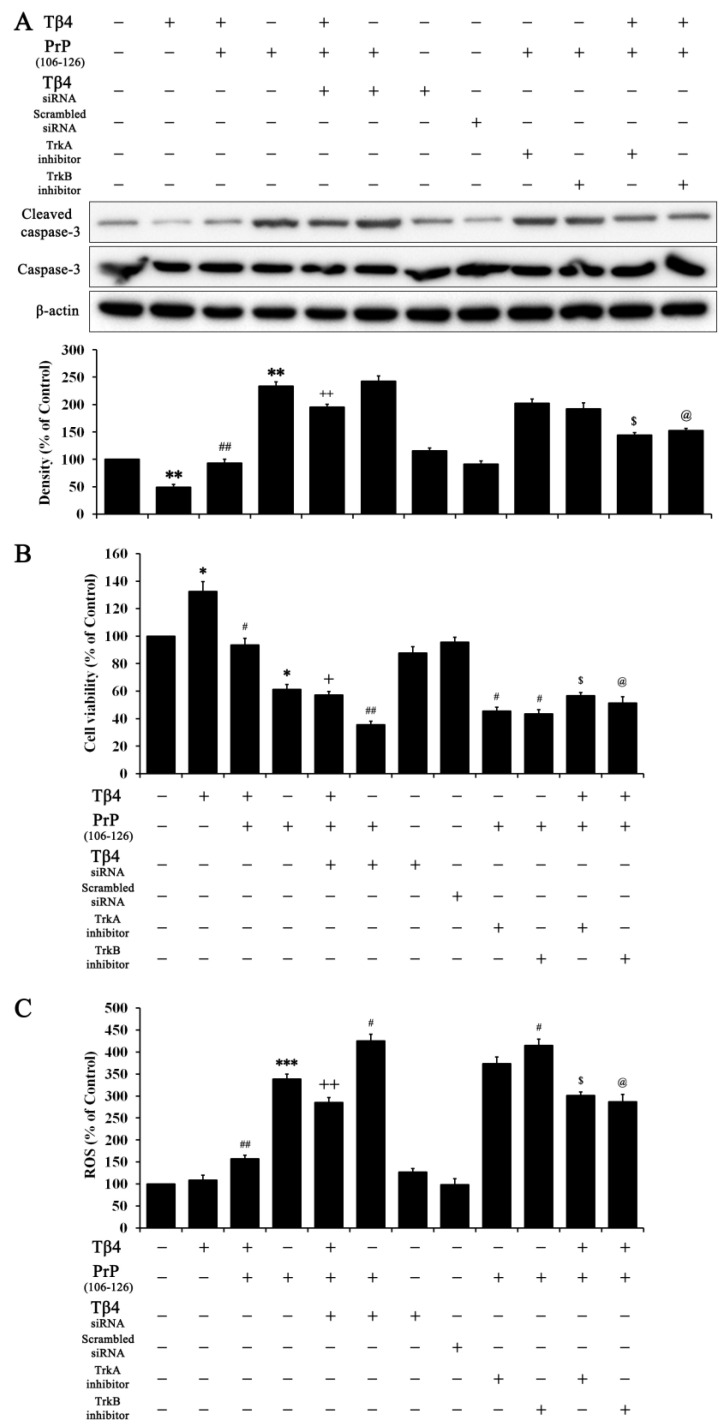
Protective effect of Tβ_4_ via neurotrophic factors and their receptors. Tβ_4_ at 400 ng/mL treated with or without scrambled siRNA, Tβ_4_ siRNA, 5 μM TrkA inhibitor, 20 μM TrkB inhibitor, and PrP (106–126) at 100 μM for 24 h. (**A**) Protein levels of cleaved caspase-3, caspase-3, and β-actin were confirmed. (**B**) Cell viability and (**C**) ROS activity were also confirmed. Data are represented as mean ± SEM (n = 3). ** p* < 0.05, compared with control. *** p* < 0.01, compared with control. **** p* < 0.001, compared with control. *# p* < 0.05, compared with only prp-treated group. *## p* < 0.01, compared with only PrP-treated group. + *p* < 0.05, compared with Tβ_4_ + PrP-treated group. ++ *p* < 0.01, compared with Tβ_4_ + PrP-treated group. $ *p* < 0.05, compared with PrP + TrkA IH-treated group. @ *p* < 0.05, compared with PrP + TrkB IH-treated group.

**Figure 7 molecules-28-03920-f007:**
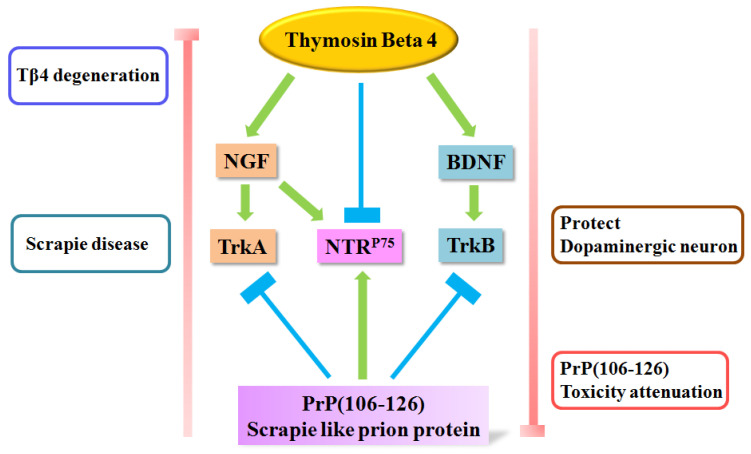
Scheme of Tβ_4_ signaling pathway on PrP (106–126)-treated cells.

## Data Availability

Not applicable.

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
