# Peer review of "Thymosin Beta 4 Protects Hippocampal Neuronal Cells against PrP (106–126) via Neurotrophic Factor Signaling"

_molecules, 2023, doi:10.3390/molecules28093920_

Round 1
Reviewer 1 Report
In the present study, the authors showed that thymosin beta 4 (Tβ4) has a protective effect against neurotoxicity induced by synthetic Prion protein peptide (PrP) (106-126) in hippocampal neuronal cells. The study is well-designed, and the results are presented in a very clear way. Also, I must say that the western blot analyses were performed very well, and the bands are quite convincing.
1. On the other hand, there are some questions that should be answered before the further publication process:
2. In Figure 1a, Tβ4 application significantly increased cell viability. Is there any possibility that Tβ4 might cause carcinogenicity in the hippocampal neuronal cells?
3. Genotoxicity analyses of Tβ4 exposure can be performed on the cell culture, such as micronucleus or chromosomal aberration assays. Or biosafety of Tβ4 should be discussed to confirm its usability.
4. There are no in vivo experiments to show the effectiveness of Tβ4. This case should be mentioned in the conclusion part.
5. Is there any information in the literature related to the blood-brain barrier transport efficiency of Tβ4? This should be indicated in the introduction part.
Author Response
On behalf of the authors, I would like to thank you for providing us the opportunity to improve our manuscript once again. We appreciated the careful reading of our manuscript as well as commenting and suggesting for better manuscript.
- On the other hand, there are some question that should be answered before the further publication process:
Response: We have carefully rewritten and reorganized our manuscript according to the comments from you. We hope that you agree with our manuscript that has been not only through revised but also strengthened by your comments. Thank you for your kind consideration
- In Figure 1a, Tβ4 application significantly increased cell viability. Is there any possibility that Tβ4 might cause carcinogenicity in hippocampal neuronal cells?
Response: According to the references and previous studies, Tβ4 has been shown to protect cells from damage and promote cells regeneration. In Figure 1A, the increased level of cell viability suggests a low potential for carcinogenicity of hippocampal cells, while in Figure 1C, data confirming the protective effect of Tβ4 against cell damage induced by PrP are shown. These results demonstrate that Tβ4 protects against PrP (106-126)-induced neurotoxicity.
This is relevant references:
- Han, H.J.; Kim, S.; Kwon J. Thymosin beta 4-induced Autophagy Increases Cholinergic Signaling in PrP (106-126)-Treated HT22 Cells. Neuroto Res 2018, 36, 58-65.
- Yang, H.; Cheng, X.; Yao, Q.; Li, J.; Ju, G. The Promotive Effects of Thymosin β4 on Neuronal Survival and Neurite Outgrowth by Upregulating L1 Expression. Neurochem Res 2008, 33, 2269-2280.
- Su, L.; Kong, X.; Loo, S.; Gao, Y.; Liu, B.; Su, X.; Dalan, R.; Ma, J.; Ye, L. Thymosin beta-3 improves endothelial function and reparative potency of diabetic endothelial cells differentiated from patient induced pluripotent stem cells. Stem Cell Research & Therapy 2022, 13, 13.
- Genotoxicity analyses of Tβ4 exposure can be performed on the cell culture, such as micronucleus or chromosomal aberration assays. Or biosafety of Tβ4 should be discussed to confirm its usability.
Response: Thymosin beta 4 was initially isolated from calf thymus, and the expression of thymosin beta in developing brain was reported in the Journal of Molecular Neurosciences in 1990. In addition, according to a report in Nature in 2004, it was found that thymosin beta 4 promotes myocardial cell migration, survival, recovery, and helps to improve heart function. Through numerous studies related to thymosin beta 4, it is considered that sufficient discussion on biological safety of thymosin beta 4 has been conducted. Relevant references are attached:
- Lin, S.C.; Morrison-Bogorad, M. Developmental expression of mRNAs encoding thymosin β4 and β10 in rat brain and other tissues. J Mol Neurosci 1990, 2, 35-44.
- Sosne, G.; Szliter, E.A.; Barrett, R.; Kernacki, K.A.; Kleinman, H.; Hazlett, L.D. Thymosin beta 4 Promotes Corneal Wound Healing and Decreases Inflammation in Vivo Following Alkali Injury. Exp Eye Res 2002, 74, 293-299.
- Bock-Marquette, I.; Saxena, A.; White, M.D.; Dimaio, J.M.; Srivastava, D. Thymosin beta4 activates integrin-linked kinase and promotes cardiac cell migration, survival and cardiac repair. Nature 2004, 432, 466-472.
- Crockford, D. Development of thymosin beta4 for treatment of patients with ischemic heart disease. Ann N Y Acad Sci 2007, 1112, 385-395.
- There are no in vivo experiments to show the effectiveness of Tβ4. This case should be mentioned in the conclusion part.
Response: We added the changes you pointed out to the conclusion section. We highlighted all changes using the “Track Changes” function in Microsoft Word in the revised manuscript.
- Is there any information in the literature related to the blood-brain barrier transport efficiency of Tβ4? This should be indicated in the introduction part.
Response: We added the information regarding the points your mentioned. Please review and confirm, and also we highlighted all changes using the “Track Changes” function in Microsoft Word in the revised manuscript.
Reviewer 2 Report
The article sent for the review entitled: Thymosin beta 4 Protects Hippocampal Neuronal Cells Against PrP (106-126) via Neurotrophic Factor Signaling has taken into consideration the important problem of the agging society i.e.: neurodegenerative disorders. One of the most investigated is prion-related ones. The PrP has shown the toxicological properties toward brain cells. Authors investigated Thymosin beta 4 as the promising factor against PrP. Their studies based on the synthetic PrP, cell culture (neuronal) studies which is not an easy challenge, small interfering RNA and intracellular ROS assay however without DNA damage analysis, and immunoblotting assay (the studies justification). In their careful studies, the Authors obtained interesting and promising results of protecting the roles of TB4 against PrP (hippocamp neuronal cells) and decreasing ROS. Due to the last one, it will be a wonder if the Authors will discuss the DNA damage formation in neurons and the mechanisms of their repair. From the editorial point of view, the article is well-written and readable. Therefore I recommend the manuscript for publication. However, I expect that the Authors will put some information about DNA damage’s role in neurodegenerative disorders.
Author Response
On behalf of the authors, I would like to thank you for providing us the opportunity to improve our manuscript once again.
Thank you for your comment regarding the role of DNA damage in neurodegenerative disorders. We agree that it is an important topic to consider, and our study does touch upon it indirectly. Specifically, our study investigated the neuroprotective effects of thymosin beta 4 (Tβ4) against neurotoxicity induced by Prion protein peptide (PrP) (106-126) in hippocampal neuronal cells. Prion diseases, such as Creutzfelt-Jakob in humans, bovine spongiform encephalopathy in cattle, and scrapie in sheep, are fatal neurodegenerative diseases characterized by misfolding and aggregation of the cellular prion protein (PrPC) into the proteinase-resistant PrPSc. Our study adds to the existing literature by providing in vitro evidence supporting the effectiveness of Tβ4 in protecting against PrP-induced neurotoxicity.
In addition, the review of the literature reveals that previous studies have reported an association between neurodegenerative disorders and DNA damage. For example, OGG1−/− MUTYH−/− mice inoculated with a prion strain exhibited reduced neuroprotection, and reduced DNA integrity in the brain, suggesting that OGG1- and MUTYH-induced BER provides neuroprotection. Furthermore, NEIL3 and NEIL2, which are DNA glycosylases that repair oxidative DNA damage, have also been investigated in the context of prion diseases, with NEIL2 shown to contribute to survival during prion disease in mice.
Overall, while our study did not directly investigate the role of DNA damage in neurodegenerative disorders, it adds to the growing body of literature suggesting that neuroprotective agents like Tβ4 may have therapeutic potential in the treatment of these diseases, and further research is needed to fully understand the mechanisms involved.
Thank you for your kind consideration
REF: de Sousa, M.M.L.; Ye, J.; Luna, L.; Hildrestrand, G.; Bjørås, K.; Scheffler, K.; Bjørås, M. Impact of Oxidative DNA Damage and the Role of DNA Glycosylases in Neurological Dysfunction. Int. J. Mol. Sci. 2021, 22, 12924. https://doi.org/10.3390/ijms222312924
Reviewer 3 Report
The article “Thymosin beta 4 Protects Hippocampal Neuronal Cells Against PrP (106-126) via Neurotrophic Factor Signaling” provides evidence that thymosin beta 4 (Tβ4) has a protective effect against neurotoxicity induced by synthetic Prion protein peptide (PrP) (106-126) in hippocampal neuronal cells. The experimental design and methods are limited but able to support the conclusions. Therefore, the manuscript can be accepted after minor revision. Please incorporate the in vivo fate of the study. Please incorporate more details about PrP (106-126).
Author Response
On behalf of the authors, I would like to thank you for providing us the opportunity to improve our manuscript once again. We have carefully rewritten and reorganized our manuscript according to the comments from you. We hope that you agree with our manuscript that has been not only through revised but also strengthened by your comments. We highlighted all changes using the “Track Changes” function in Microsoft Word in the revised manuscript. Thank you for your kind consideration